

# Turkish nursing students' compliance to standard precautions during the COVID-19 pandemic

Sevcan Topçu and Zuhal Emlek Sert

Ege University, İzmir, Turkey

## ABSTRACT

**Objectives**. The aim of this study was to assess nursing students' compliance to standard precautions during the COVID-19 pandemic.

**Methods**. A cross-sectional study was conducted from December 2021 to June 2022, 816 nursing students participated in the study. A socio-demographic questionnaire and Compliance with Standard Precautions Scale were used to collect data. Means and percentages were used to report socio-demographic characteristics multiple regression analysis used to identify the factors influencing compliance with standard precautions.

**Results**. The mean age of nursing students was 21.30 ± 1.31 years. The majority of nursing students were female, with 703 (86.2%) being female and 113 (13.8%) being male. Compliance among nursing students was 76.8% overall. Nursing students reported the highest compliance (97.7%) with putting used sharp articles into sharp boxes, with 97.1% compliance for covering mouth and nose when wearing a mask. Participants reported the lowest (38.6%) when it came to not recapping used needles after giving an injection. Regression analysis revealed that gender, year of study, and having needlestick injury or contact with blood/body fluids experience all influenced nursing students' compliance with standard precautions.

**Conclusions**. During the pandemic, nursing students compliance to standard precautions was optimal, according to this study. More research should be done to assess nursing students' compliance with standard precautions and the effect of infection control strategies used to prevent COVID-19 transmission.

Corresponding author
Sevcan Topçu, sevcan.topcu@hotmail.com

## INTRODUCTION

Healthcare-associated infections (HAIs) are infections contracted while undergoing treatment for another condition (*ECDC, 2022*; *HHS, 2021*). HAIs not only endanger the patients' health and lives, but they also impose additional emotional and financial burdens on patients, their families and the healthcare system, such as direct economic loss and prolonged hospitalization (*HHS, 2021*; *Jia et al., 2019*). While HAIs primarily affect patients, they also affect healthcare workers (HCWs) and students who are regularly exposed to bacteria, fungi, viruses, or other pathogens that cause HAIs (*Brosio et al., 2017*). Respiratory tract infections, surgical site infections, urinary tract infections, bloodstream infections, and gastrointestinal infections are the most commonly reported types of HAIs
(*ECDC, 2022*). The ECDC reported 8.9 million HAIs in European hospitals and long-term care facilities in 2018 (*ECDC, 2018*). They estimated that one out of every 15 patients in hospitals and one out of every 26 patients in long-term care facilities had HAIs (*ECDC, 2018*).

COVID-19 first appeared in in December 2019 in Wuhan, China, (*WHO, 2020a*). It quickly spread throughout the world, and the World Health Organization (WHO) declared the COVID-19 outbreak a global pandemic in March 2020 (*WHO, 2020a*). The rapid spread of COVID-19 resulted high hospitalization rates, necessitating an increase in total hospital capacity (*McCabe et al., 2020*). The efforts of healthcare facilities to quickly adapt to increased hospital demand influenced HAI rates (*World Health Organization & Regional Office for Europe, 2020b*). According to the Centers for Disease Control and Prevention (CDC), the incidence of central line-associated bloodstream infections (CLABSIs), catheter-associated urinary tract infections (CAUTIs), ventilator-associated events (VAEs), and methicillin-resistant *Staphylococcus aureus* (MRSA) bacteremia in 2021 will be significantly higher than in 2019 (*CDC, 2022*; *Weiner-Lastinger et al., 2022*). Also, among all HAIs types, VAEs experienced the greatest increase in the third quarter of 2021, when the Delta variant drove COVID-19-related hospitalizations to all-time highs (*CDC, 2022*; *Weiner-Lastinger et al., 2022*). COVID-19, on the other hand, had a positive effect on some HAIs, including *Clostridioides difficile* (*C. diff*) (*CDC, 2022*; *Weiner-Lastinger et al., 2022*). It was discovered that *C. diff* infections in healthcare settings have significantly decreased as a result of pandemic-related improvements such as hand hygiene, personal protective equipment (PPE) practices, and environmental cleaning (*CDC, 2022*; *Weiner-Lastinger et al., 2022*). The COVID-19 burdens on healthcare systems may also limit their ability to sustain surveillance, which is the foundation of the fight against HAIs (*Wee et al., 2021*).

Compliance with standard precautions (SPs) is the most effective way to prevent HAIs (*Arruum et al., 2021*). In 1996, the CDC changed the infection control practice from Universal Precautions to Standard Precautions. SPs are based on the premise that all blood, body fluids, secretions, excretions (except sweat), nonintact skin, and mucous membranes may contain infectious agents that are transmissible. SPs are infection-prevention practices that apply to all patient care, regardless of whether the patient is suspected or confirmed to have infection, in any setting where health care is provided (*CDC, 2018*). SPs aim to protect both health workers and patients by lowering the risk of microorganism transmission from both recognized and unknown sources (*WHO, 2022*). SPs, when followed consistently, can prevent the spread of microorganisms between patients, health workers, and the environment. Risk assessment, hand hygiene, respiratory hygiene and cough etiquette, patient placement, PPE, aseptic technique, safe injections and sharp injury prevention, environmental cleaning, handling of laundry and linen, waste management, decontamination and reprocessing of reusable patient care items and equipment, decontamination of spilled wastes, proper disposal, and the prevention of cross infection are all important components of SPs (*CDC, 2018*; *WHO, 2022*). As a result, it is critical for HCWs to follow SPs. To prevent COVID-19 transmission in healthcare facilities during the COVID-19 pandemic, aggressive infection prevention and control (IPC) measures such as the use of PPE (especially masks), droplet and contact

isolation precautions for patients with respiratory symptoms, and visitor restrictions were implemented. Because of concerns about cost-effectiveness and sustainability prior to the COVID-19, such strategies were used in high-risk units such as intensive care units and infection clinics (*Wee et al., 2021*). Attention to traditional HAIs prevention programs and long-established infection control measures among HCWs decreased during COVID-19 due to increased total hospital capacity and hospitalization rates, as well as controlled personnel availability. (*Weiner-Lastinger et al., 2022*; *Deiana et al., 2022*). Due to a shortage of nurses during the pandemic, nursing students (NSs) were among the frontline HCWs in healthcare facilities, particularly COVID-19 clinics (*Yılmaz, Karaman & Yılmaz, 2021*). NSs, as members of the healthcare team, play a critical role in the prevention of HAIs because they treat and care for a diverse range of patients during their clinical placement. Many studies were conducted prior to the pandemic to assess NSs compliance with SPs. According to these studies, NSs' compliance with SPs was low or moderate (*Bouchoucha et al., 2021*; *Cruz, 2019*; *Moon, Hyeon & Lim, 2019*). Nonetheless, evidence for the effect of COVID-19 on NSs' compliance with SPs is limited. As a result, the purpose of this study was to assess NSs' compliance with SPs during the COVID-19.

Study questions:

- What is the NSs' compliance with standard precautions during COVID-19?
- Is there a difference in compliance with SPs among NSs based on key sample characteristics such as gender, year of study, needlestick injury, or contact with blood/body fluids?

## MATERIALS & METHODS

### Study design and sampling

From December 2021 to June 2022, a cross-sectional survey was conducted at a nursing school in Turkey. This study's population consisted of those with a Bachelor of Science in NSs who had already begun their practical training period. The study employed convenience sampling. The study's sample included 925 second-, third-, and fourth-year NSs. The study's inclusion criteria were (1) voluntary study participation and (2) being in the practical training period.

Because of their lack of clinical experience, first-year NSs were barred from participating. This study included all NSs who met the inclusion criteria.

### Data collection

Before each class's teaching session, researchers explained the purpose and procedure of the study and invited NSs in their second, third, and fourth years to participate in the paper-based questionnaire survey. The paper survey was distributed to NSs who agreed to participate in the study. The questionnaire was distributed to 910 second-year students. The study included 816 NSs who completed a self-administered the survey. Researchers assured NSs that their participation in the survey would be entirely voluntary and confidential, and that it would have no bearing on their grades.

## Instruments

The socio-demographic questionnaire and Compliance with Standard Precautions Scale (CSPS) were used to collect data (*Lam, 2011*). The socio-demographic uestionnaire included four questions: age, gender, year of study, needlestick injury, or contact with blood/body fluids.

The CSPS is a self-reported compliance with SPs among nurses and NSs based on international preventive measures developed by the Centers for Disease Control and Prevention and the World Health Organization (*Lam, 2011*). Lam developed the CSPS by modifying the Universal Precautions Scale because SPs combine the major features of Universal Precautions and Body Substance Isolation. The CSPS assesses compliance with the major dimensions of SPs and was comprehensively designed to describe the daily routine of nurses or NSs in performing infection control practices in their work (*Cruz et al., 2016*; *Lam, 2011*). CSPS consist of 20 items, and five areas that include use of protective devices (or PPE) (six items), disposal of sharps (three items), disposal of waste (three items), decontamination of spills and used articles (one item), and prevention of cross-infection from person to person (seven items). The response set is a 4-point Likert-scale with responses like "never," "seldom," "sometimes," and "always." With the majority of the items being positively worded statements, except for questions 2, 4, 6, and 15. Only the "always" option in positively worded statements and the "never" option in negatively worded statements receive a score of 1. The other options receive no points. The total scores range from 0 to 20, with a higher score indicating greater compliance with SPs. The compliance rate is th percentage of average compliance with all 20 items (*Lam, 2014*). *Samur, Seren Intepeler & Lam (2020)* conducted a Turkish validity and reliability study with 411 nurses. In this study, the Turkish version CSPS was used. The Turkish version of the CSPS demonstrated good internal consistency and reliability (Cronbach's $\alpha$, 0.71; intraclass correlation coefficient, 0.84). The content validity of the scake was 0.99. The authors have permission from the copyright holders to use this instrument.

## Data analyses

The Statistical Package for the Social Sciences, version 21 (SPSS, Inc., Chicago, IL, USA) was used to analyze the data. For the sociodemographic variables and NSs' compliance with SPs, descriptive statistics (mean, percentage, etc.) were used. Multiple regression analysis was used to identify key independent variables using the compliance rate with SPs as the dependent variable. As independent variables, gender, year of study, needlestick injury, or contact with blood/body fluids were identified. For the detection of multicollinearity, values for the variance inflation factor ($>10$) and tolerance level ($<0.1$) were used (*Dormann et al., 2013*). For all effects, we used the standard significance level of $\alpha = 0.05$.

## Ethical considerations

To conduct the study, permission was obtained from Ege University Scientific Research and Publication Ethics Committee (Approval Number: 13/09-1213). Before data collection, participants were informed about the research objectives and procedures, and their written permission was obtained *via* an informed consent form. All participants were informed

that their participation in the study entirely voluntary and would have no bearing on their grades.

## RESULTS

The mean age of NSs' was 21.30 ± 1.31 years (range = 19–31). The majority of NSs were female, with 703 (86.2%) being female and 113 (13.8%) being male. Of NSs, 311 (38.1%) were second-year students, 274 (33.6%) were third-year students and 231 (28.3%) were fourth-year students. Only 8.6% of NSs experienced needlestick injuries or contact with blood/body fluids (Table 1).

Among NSs, the overall of compliance rate was 76.8%. The NSs reported the highest compliance with putting used sharp articles into sharp boxes (97.7%), with 97.1% compliance for covering mouth and nose when wearing a mask, and 93.0% compliance for wearing gloves when exposed to body fluids, blood products and any excretion of patients (Table 2). The NSs reported the lowest compliance in the not recapping used needles after giving an injection (38.6%), followed by using alcoholic hand rubs as an alternative if hands are not visibly soiled (55.4%),

A multiple regression analysis was used to identify variables that were independently related to NSs' compliance with SPs (Table 3). The model was statistically significant ($F = 77.75$, $p < 0.05$). It accounted for roughly 28% of the variance in SPs compliance. All of the independent variables were significantly associated with more favorable compliance with SPs ($p = 0.01$). Having needlestick injury or contact with blood/body fluids experience had the highest relative effect, as indicated by the standardized beta value ($\beta = -0.40$). Gender ($\beta = -0.26$) and year of study (being third year $\beta = -0.12$; being fourth year $\beta = -0.19$) were other independent variables associated with SP compliance. Being female, being in your second year of nursing school, and not having had a needlestick injury or contact with blood/body fluids were all associated with higher compliance with SPs.

## DISCUSSION

The study assessed how COVID-19 affected NSs' compliance with SPs. During the pandemic, the overall rate of compliance with SPs among NSs was 76.8%. *Colet et al. (2017)* and *Alshammari et al. (2018)* used CSPS to assess compliance with SPs among Saudi NSs. They (2021) dicovered that prior to the pandemic, the overall compliance rate with SPs was 61% and 60.1%, respectively. *Bouchoucha et al. (2021)* used CSPS to assess compliance with SPs among Australian NSs. In a nonpandemic environment, they reported that Australian NSs had good compliance with SPs (68.9%). During the COVID-19 pandemic, *Rizk & Siam (2021)* assessed the effect of the telenursing education program on Egyptian nurses' compliance with SPs using the same measurement tool (CSPS). Prior to telenursing education, the overall compliance rate with SPs among nurses was 66.2%. *Albaqawi et al. (2020)* used a different measurement tool to assess COVID-19 prevention behaviors among Saudi NSs. They found that NSs' used COVID-19 prevention behaviors effectively. In terms of health systems, Turkey shares similarities with Egypt and Saudi Arabia, and the majority of health services are provided by the government. Furthermore, Turkey, Egypt, and

**Table 1  Characteristics of nursing students.**

| Characteristics ($N = 816$) | | n | % |
|---|---|---|---|
| **Age (Mean $\pm$ SD)** | $21.30 \pm 1.31$ | | |
| **Gender** | Female | 703 | 86.2 |
| | Male | 113 | 13.8 |
| **Year of Study** | Second year | 311 | 38.1 |
| | Third year | 274 | 33.6 |
| | Fourth year | 231 | 28.3 |
| **Needlestick injury or contact with blood/body fluids** | Yes | 70 | 8.6 |
| | No | 746 | 91.4 |

Saudi Arabia are Muslim countries. However, Australia differs from Turkey in terms of healthcare financing and the religious structure of the society. *WHO (2020a)* recommended key IPC measures for health facilities during COVID-19, including respiratory etiquette and hand hygiene best practices; contact, droplet, and airborne precautions; adequate environmental cleaning and disinfection; ensuring adequate ventilation; isolation facilities for COVID-19 patients; visitor restrictions; and, where possible, maintaining a physical distance of at least 1 meter among all individuals in health facilities, especially in indoor settings. These measures have a similar structure to the SPs used to prevent HAIs. Prior to clinical placements during the pandemic, nursing schools and training hospitals routinely provided students with training on infection control measures in COVID-19. Furthermore, to prevent COVID-19 transmission, clinical nurse educators carefully followed up on IPC strategies observed NSs in clinical placements during the pandemic. The implementation of these key IPC pandemic measures has been effective in reducing HAIs such as *C. diff* (*CDC, 2022*; *Lastinger et al., 2022*). However, increased pandemic-related IPC measures have resulted in an increase in HAIs such as CLABSIs, CAUTIs, and VAEs during COVID-19 (*Baker et al., 2022*; *CDC, 2022*; *Lastinger et al., 2022*). The compliance rate of Turkish NSs with SPs during the pandemic was relatively high when compared to previous studies conducted prior to the pandemic that uses the same measurement tool. The findings of this study indicate that IPC strategies used to prevent COVID-19 transmission have a positive effect of on the compliance with SPs among NSs.

In this study, NSs reported 78.7% compliance with PPE use. According to current evidence, COVID-19 spreads primarily through respiratory droplets, but it may also spread through contact with contaminated surfaces. As a result, WHO advised HCWs to to wear masks continuously in areas where there is a patient with suspected or confirmed COVID-19 and to use gloves when necessary (*WHO, 2021*). In this study, NSs reported the highest compliance with covering their mouth and nose when wearing a mask (97.1%), as well as wearing gloves when exposed to body fluids, blood products and any excretion of patients (93%). Many studies conducted prior to the pandemic found that, NSs wore gloves with high compliance (*Bouchoucha et al., 2021*; *Donati et al., 2019*). However, when compared to this study, most studies found that NSs were less likely to cover their mouth and nose when wearing a mask (*Alshammari et al., 2018*; *Bouchoucha et al., 2021*; *Colet et al., 2017*). Another interesting finding from this study is that 31.9% of NSs reuse a surgical

**Table 2** Compliance of nursing students with standard precautions ($N = 816$).

| Items | Compliance rate % |
|---|---|
| **Use of protective device** | **78.7** |
| Q7. I remove Personal Protective Equipment (PPE) in a designated area. | 78.7 |
| Q10. I wear gloves when I am exposed to body fluids, blood products and any excretion of patients. | 93.0 |
| Q13. I wear a surgical mask alone or in combination with goggles, face shield and apron whenever there is a possibility of a splash or splatter. | 68.0 |
| Q14. My mouth and nose are covered when I wear a mask. | 97.1 |
| Q15. I reuse a surgical mask or disposable PPE | 69.1 |
| Q16. I wear a gown or apron when exposed to blood, body fluids or any patient excretions. | 66.4 |
| **Disposal of sharps** | **66.5** |
| Q4. I recap used needles after giving an injection. | 38.6 |
| Q5. I put used sharp articles into sharps boxes | 97.7 |
| Q6. The sharps box is disposed only when it is full. | 63.1 |
| **Disposal of waste** | **81.5** |
| Q17. Waste contaminated with blood, body fluids, secretion and excretion is placed in red plastic bag s irrespective of the patient's infection status. | 81.5 |
| **Decontamination of spills and used article** | **85.0** |
| Q18. I decontaminate surfaces and equipment after use. | 91.4 |
| Q19. I wear gloves to decontaminate used equipment with visible soils. | 86.6 |
| Q20. I clean up spillage of blood or other body fluids immediately with disinfectants | 77.0 |
| **Prevention of cross infection from person to person** | **75.5** |
| Q1.I wash my hands between patient contacts. | 84.4 |
| Q2. I only use water for hand washing | 74.3 |
| Q3. I use alcoholic hand rubs as an alternative if my hands are not visibly soiled | 55.4 |
| Q8. I take a shower in case of extensive splashing even after I have put on PPE. | 60.5 |
| Q9. I cover my wound(s) or lesion(s) with waterproof dressing before patient contacts. | 77.8 |
| Q11. I change gloves between patient contacts | 85.2 |
| Q12. I decontaminate my hands immediately after removal of gloves. | 91.1 |
| **CSPS** | **76.8** |

mask or disposable PPE. This could be due to a lack of PPE in clinical settings during the pandemic.

Although NSs were highly compliant with disposing of used sharp articles into sharp boxes, they were the least compliant with not recapping used needles after injections. Capturing used needles is the leading cause of needlestick injuries among NSs in our country

**Table 3  Factors influencing nursing students' compliance with SPs ($N = 816$).**

| Predictor variable | Unstandardized coefficients | | Standardized coefficients | | | CI | |
|---|---|---|---|---|---|---|---|
| | B | SE | $\beta$ | t | p | Lower bound | Upper bound |
| Gender (reference group:female) | −17.30 | 2.01 | −0.26 | −8.62 | 0.01 | −21.24 | −13.36 |
| Year of study (reference group: second-year) | | | | | | | |
| Third year | −5.66 | 1.61 | −0.12 | −3.51 | 0.01 | −8.83 | −2.49 |
| Fourth year (İnternship) | −9.71 | 1.72 | −0.19 | −5.65 | 0.01 | −13.01 | −6.34 |
| Needlesstick injury or contact with blood/body fluids (reference group:no) | −32.93 | 2.45 | −0.40 | −13.42 | 0.01 | −37.74 | −28.11 |
| $R = 0.53$, $R^2 = 0.28$, $F = 77.75$, $p = 0.01$ | | | | | | | |

**Notes.**
B is the unstandardized coefficients; SE is the standard error; $\beta$ is standardized beta; p is level of significance as $p < 0.05$; CI is confidence interval.

(*Kepenek & Şahin Eker, 2017*). *Colet et al. (2017)* found that 49.2% of NSs in a cross-sectional study with NSs in Saudi Arabia did not recap used needles after administering an injection. According to *Bouchoucha et al. (2021)*, NSs have a 75.1% compliance rate with not recapping used needles. In a study of Ghanaian NSs during the pandemic, 21.4% reported never recapping used needles (*Balegha, Yidana & Abiiro, 2021*). Furthermore, during the pandemic, NSs' compliance with not recapping used needles decreased. This outcome could be attributed to a decreased emphasis on other prevention strategies as a result of increased pandemic-related IPC measures. Low compliance among NSs for recapping used needles is undesirable because it may lead to an increase in the number of occupational accidents.

NSs reported high compliance with waste disposal and decontamination of spills and used article. 91.4% of those polled said they always decontaminated surfaces and equipment after use and 86.6% reported they wore gloves to decontaminate used equipment with visible soils. However, the compliance rate for immediately cleaning up spilled blood or other bodily fluids with disinfectants is 77.0%. A previous study of Saudi NSs found that wearing gloves to decontaminate used equipment with visible soils (64.8%), decontaminating surfaces and equipment after use (52.5%), and immediately cleaning up spilled blood or other body fluids with disinfectants (62.3%) were all low compliance measures (*Colet et al., 2017*). A study with Australian NSs found that compliance with cleaning up spilled blood or other body fluids immediately with disinfectants (86%) was high, compliance with decontaminating surfaces and equipment after use (78.5) was low, and compliance with wearing gloves to decontaminate used equipment with visible soils (88.2%) was low (*Bouchoucha et al., 2021*). These findings revealed that compliance with waste disposal and decontamination of spills and used article among NSs varies from country to country and culture to culture regardless of whether the pandemic occurred before or after.

Participants reported 75.5% compliance with strategies to prevent person to person cross-infection, with 91.1% compliance for decontaminating hands immediately after removing of gloves, 85% compliance for changing gloves between patient contacts and 84% compliance for washing hand between patient contacts. Only 55.4% of participants

reported they use alcoholic hand rubs as an alternative if their hands are not visibly soiled. According to the findings of this study, compliance with the prevention of cross infection from person to person is quite high when compared with to studies conducted prior to the pandemic (*Alshammari et al., 2018*; *Colet et al., 2017*; *Moon, Hyeon & Lim, 2019*). In the current study, NSs reported low compliance with using alcoholic hand rubs as an alternative if my hands were not visibly soiled. Hand washing and disinfection were recommended by *WHO (2021)* as the most effective protective measures against the COVID-19 pandemic. Hand hygiene and disinfectant use among HCWs increased at the start of the pandemic, but returned to prepandemic levels over time (*Sandbøl et al., 2022*; *Stangerup et al., 2021*). The study was conducted in the first clinical placements of NSs during the COVID-19 pandemic, which may explain high compliance with hand washing among NSs As a result, compliance with hand hygiene among NSs should be re-evaluated following the pandemic.

Female NSs had higher compliance with SPs than male NSs, according to our finding. This gender gap in compliance with SPs was also observed in Saudi Arabia and the Croatia, where female students outperformed male students (*Alshammari et al., 2018*; *Colet et al., 2017*; *Mestrovic, Neuberg & Kozina, 2020*). *Cruz (2019)* found that male NSs were more likely to comply with SPs than female. Gender differences in SPs compliance with should be taken seriously in order to understand why these differences occur and, eventually, to implement the necessary measures to improve compliance. As a result, qualitative studies to investigate the possible causal relationship between gender and compliance with SPs are suggested. This study found that second-years NSs had higher compliance with SPs than third years and fourth years. Similarly, previous studies reported that intern NSs had low compliance with SPs (*Cruz, 2019*; *Alshammari et al., 2018*; *Colet et al., 2017*). These findings imply that students' exposure to clinical settings increases, so does their compliance with SPs. The duration of their involvement in clinical placements increases as the year of study among NSs participating in our study increases, and students, particularly fourth-year NSs, take responsibility like nurses in clinical placements and encounter more complex situations. These circumstances may cause NSs' compliance with SPs to wane as the study year progresses. The current study found that NSs who had needlestick injuries or had contact with blood/body fluids had low compliance with SPs. Similarly, *Moon, Hyeon & Lim (2019)* reported that compliance with SPs is lower among NSs who have experienced needlestick injury. According to this study, having a needlestick injury or contact with blood/body fluids were predictors of compliance with SPs. Some SPs (such as recapping used needles after injections, storing used sharps in sharps boxes, and so on) are also effective in preventing needlestick injuries, which can lead to serious or fatal infections with bloodborne pathogens such as hepatitis B, hepatitis C, or HIV (*NIOSH, 2021*). Strict compliance to SPs can reduce exposure to blood/body fluids and needlestick injuries. These findings highlight the importance of SPs in both preventing needlestick injuries and HAIs.

## Limitations

While this study contributed valuable findings to the literature, it did have some limitations. A convenience sample of NSs from one nursing school was used to collect data. As a result,

the generalizability of results may be limited. Furthermore, the CSPS is a self-reported questionnaire. This can lead to social desirability bias in respondents.

## CONCLUSIONS

The main strategy for preventing health-related infections is compliance with SPs. Strict compliance to SPs is required to ensure the safety of patients and HCWs. According to this study, the compliance with SPs was relatively higher among Turkish NSs during the pandemic. Compliance with SPs was high among NSs in terms of decontamination of spills and used articles, as well as waste disposal. However, there were obvious issues with sharp disposal compliance among NSs. Gender, study year, and previous needlestick injury or contact with blood/body fluids were all predictors of compliance with SPs. Males and NSs who had a needlestick injury or had contact with blood/body fluids had lower compliance with SPs than others. It is critical to cultivate and sustain a supportive culture of SPs compliance among NSs. A culture of strict compliance to SPs should be supported by training hospitals, nursing schools, and clinical nurse educators. It is recommended that strategies similar to those hospitals to prevent COVID-19 transmission during the pandemic (training, follow-up, observation, *etc.*) be used to increase compliance with SPs. More research is needed to determine the impact of the pandemic on NS compliance with SPs.

## ACKNOWLEDGEMENTS

The authors would like to thank everyone who helped with this study. We are grateful to Ege University Planning and Monitoring Coordination of Organizational Development and Directorate of Library and Documantation for their support in editing and proofreading service of this study.

### Funding
The authors received no funding for this work.

### Competing Interests
The authors declare there are no competing interests.

### Author Contributions
- Sevcan Topçu conceived and designed the experiments, performed the experiments, analyzed the data, prepared figures and/or tables, authored or reviewed drafts of the article, and approved the final draft.
- Zuhal Emlek Sert conceived and designed the experiments, performed the experiments, analyzed the data, prepared figures and/or tables, authored or reviewed drafts of the article, and approved the final draft.

## Human Ethics

The following information was supplied relating to ethical approvals (i.e., approving body and any reference numbers):

Ege University Scientific Research and Publication Ethics Committee granted Ethical approval to carry out the study within its facilities (Approval Number: 13/09-1213).

## Data Availability

The data is available in the Supplemental File.

## Supplemental Information

Supplemental information for this article can be found online at http://dx.doi.org/10.7717/peerj.15056#supplemental-information.

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
