# Peer review of "Turkish nursing students' compliance to standard precautions during the COVID-19 pandemic"

_PeerJ, doi:10.7717/peerj.15056_

## Round 0.1 · original submission · Major Revisions

Dear Authors, thank you for submitting your manuscript to PeerJ. This article has been reviewed by relevant experts from the field. They found the manuscript carries merits, but raised several concerns which should be addressed before considering the manuscript for publication. Please refer to their comments, more specifically on the methodology, presentation of results, and limitations of this study.

Moreover, this manuscript needs your attention to improve scientific writing, grammar, syntax, and spelling. I would suggest to opt an editing service or considering assistance from a proficient English speaker.

Reviewer 1 ·

Basic reporting

This is interesting article. Research background and motivation were clearly stated. The data collection approach and statistical analysis were used appropriately and described clearly in the manuscript.

1. One comment on the results of the multiple regression analysis. The results section only reported the overall model fit for the model and the variance explained by the model, and mentioned the statistically significant predictors in the model, but did not clearly quantify the effect of predictors on the outcome or provide specific interpretation on those effects of these predictors.

2. The English language should be improved to ensure that the grammars are correct and international audience can clearly understand the manuscript. Some examples where the language could be improved include lines 247, 251, ect. – the current phrasing makes comprehension difficult. I suggest the authors have a colleague who is proficient in English and familiar with the subject matter review the manuscript, or contact a professional editing service.

3. A session to discuss the limitations in the article seems missing. The authors should discuss the limitations of this study and clearly describe them in the Discussion section.

Experimental design

See comments above

Validity of the findings

See comments above

·

Basic reporting

Thanks for the opportunity to review this interesting paper. The current study is on a topic of relevance and general interest to the readership of the journal.
Generally, the study is well written and presents a clear presentation of the findings. I enjoyed the reading. The following comments were provided below, and might be helpful to improve your study:

Abstract

• Objectives. The aim of this study was to evaluate the compliance with SPS among NSS during the COVID-19 pandemic.

• Kindly spell out the abbreviation used in the objectives, as any abbreviation should be written in full, then use the abbreviation

• Refers to the study's setting and sample type.
• Instead of conducted, use conducted among
• Result: use female/ male instead of women/ men

• You did not mention the result of the regression analysis used.
• Support your conclusion with some improvement strategies you recommend for future practice or research.

Introduction:needs improvement. Consider the following comments:
• Mention the World Health Organization (WHO) completely for the first time.
• Update the old reference.
• We recommend you elaborate more on the dimension of compliance with standard precautions and add operational or conceptual definitions of the dimensions you measured in the CSPS.

Experimental design

Materials & Methods
Needs improvement. Consider the following comments:
indicate your study questions
• Indicate the type of sample and sample determination criteria.
• It is better to divide the data collection section into instruments and data collection procedures.
• Explain how did you approach the students through their academic schedule?
• Did you use hand-delivered or electronic surveys?

The CSPS

• Indicates that the CSPS was developed by whom?
• The total scores range from 0 to 20, and a higher score indicates better compliance with SPs.
• Indicate the point or the score (cutt of point) on which you judged the score as better compliance with SPs.
• Explain the Turkish validity and reliability that was conducted by Samur, Intepeler, Lam (2020).
• Data analysis. Add the level of significance and indicate dependent and independent variables in your regression analysis.

Ethical considerations
• Explain how do you protect the rights of venerable groups in your study ( students)

Validity of the findings

Clear findings,data have been provided; they are robust, statistically sound, & controlled.: Consider the following comments:
Results
• Use female/ male instead of women/ men
• A multiple regression analysis
• According to table 3, kindly explain the effect of each variable on CSPS.

Discussion
• It is preferable to support each finding with at least two studies or references.
• In the last paragraph, explain the justification and rationale of your findings and support them with your recommendation for further qualitative study

Conclusions are well stated, linked to original research question & limited to supporting results.

Additional comments

• The implications of your study for practice ,education, and research.
• Add the limitations of your study.

Add the confidence interval and level of significance to table 3.
References
Update older references prior to 2015.

Reviewer 3 ·

Basic reporting

Thank you very much for giving me the opportunity to review this manuscript. Kindly find below my suggestions to improve this manuscript.

1. The manuscript should conform to professional standards. Line 40, 183: The first time using the abbreviation, should be started by writing the full phrase and abbreviation in brackets.

2. The English language and grammar should be improved to make sure international audiences can clearly understand this study (e.g., lines 43-44, 170-171, 192-203, 243-246, etc). This includes using lowercase and uppercase letters (e.g. line 43, 193, etc.). Is the Socio-Demographic Questionnaire a proper noun?

3. Multiple regression analysis was used to identify factors influencing students' compliance. The results should also be included under the results of the abstract. For the result, line 168: p-value should be written as p<0.01) (refer to supplemental file).

4. Line 110 - Cruz (2019) not listed under references.

5. The authors provide discussion for all domains of SP. To improve the discussion and increase the comprehension of the audience, I would suggest the authors to summarize the key results based on the objective, and organize the issue based on its importance. Please state the strength and weakness of the study.
- Lines 182-188. The sentences need to be revised by making a clear relationship between the Covid-19 case and HAIs.
- Line 180, 188-190- the statement is not clear. The authors should state the citation of the studies and the practice is by who.
- Line 190-191: the statement needed to be revised, particularly “..positive effect..” because the result is not much different from the previous studies (pre COVID-19). Please also explain, how IPC strategies were implemented to the students in the study that influenced their compliance scores?
-Line 208. Need to revise the sentence when the authors generalized to “…our country..”, but the study is conducted at a single centre.
-Line 215-216, 259-260: Not clear. What are the specific recommendations?
-Line 255-256: Need explanation the reason of the results

6. I noticed that the authors frequently compared their study with several previous studies that they referred to as "pre-pandemic studies." To improve the discussion, I recommend the authors to explain briefly the study that made them qualify to be the main reference as pre-pandemic, particularly on aspects of population, country, instrument used, sample, etc. The discussion would be inaccurate if the authors relied only on the findings while healthcare setting/system, health policies, and nursing practice can be different among the countries and may influence their compliance.

Experimental design

1. The objective "to evaluate" is unambiguous and seems less suitable for this study because it is only a cross-sectional study. I suggest the authors select other terms that will accurately reflect their research methods and results, such as examine, investigate, assess, etc.

2. The research question is not clear. The knowledge gap being investigated should be properly explained. To improve, please explain the difference in 1) SP practices pre and during COVID 19, and 2) types of HAIs and the factors contributing to the increase in HAIs pre and during COVID 19. And also, please explain the relationship between COVID-19 and the need for SP compliance among the NS.

3. For methods under data analysis. To improve, I recommend the authors to state the potential confounder variables included in the multiple regression analysis and also to include the significant p-value set for this study.

Validity of the findings

Line 53-54, 263-272: The conclusion doesn't reflect the overall results and should be improved by connecting to the original research question. To revise "..the negative and positive effects..." as this is an unambiguous conclusion.

---

## Round 0.2 · accepted · Accept

Dear Authors

Thank you for considering the comments of the reviewers.

·

Basic reporting

the authors addressed the reviewer's comments and suggestions

Experimental design

the authors addressed the reviewer's comments and suggestions

Validity of the findings

the authors addressed the reviewer's comments and suggestions

Reviewer 3 ·

Basic reporting

The English is clear and unambiguous but needs proofreading, particularly related to the use of capitals/small letters (e.g. Multiple regression analysis in the abstract), spacing (e.g., in title), spelling (e.g. miss the word of ‘q’ for questionnaire in instrument- line 153, 154), the use of full stop (e.g. line 178) etc.

I noticed there are two different versions of abstracts

There is also a typo of short term for standard deviation in Table 1. Under heading of results (line 209 and 210), the P-value is not same as reported in the Table 3

Experimental design

no comment

Validity of the findings

no comment

Additional comments

no comment